# Linkages between soil carbon, soil fertility and nitrogen fixation in *Acacia senegal* plantations of varying age in Sudan

Wafa E. Abaker[1,2], Frank Berninger[3], Gustavo Saiz[4,5], Jukka Pumpanen[6] and Mike Starr[3]

[1] Department of Forest Sciences, Viikki Tropical Resources Institute, University of Helsinki, Helsinki, Finland
[2] Faculty of Forestry, University of Khartoum, Khartoum, Sudan
[3] Department of Forest Sciences, University of Helsinki, Helsinki, Finland
[4] Department of Environmental Chemistry/Faculty of Sciences, Universidad Católica de la Santísima Concepción, Concepción, Chile
[5] Department of Life Sciences, Imperial College London, Ascot, UK
[6] Department of Environmental and Biological Sciences, University of Eastern Finland, Kuopio, Finland

Corresponding author
Wafa E. Abaker,
wafa.abaker@helsinki.fi

## ABSTRACT

**Background:** Over the last decades sub-Saharan Africa has experienced severe land degradation and food security challenges linked to loss of soil fertility and soil organic matter (SOM), recurrent drought and increasing population. Although primary production in drylands is strictly limited by water availability, nutrient deficiencies, particularly of nitrogen (N) and phosphorus (P), are also considered limiting factors for plant growth. It is known that SOM (often measured as soil organic carbon (SOC)) is a key indicator of soil fertility, therefore, management practices that increase SOM contents, such as increasing tree cover, can be expected to improve soil fertility. The objectives of this study were to investigate the effect of *Acacia senegal* (*Senegalia senegal*) trees on soil nitrogen, phosphorus and potassium (K) in relation to SOC, the potential of *A. senegal* for $N_2$ fixation, and to identify possible N and P ecosystem limitations.

**Methods:** Soil nutrient (total N, P, K and available P and exchangeable K) concentrations and stocks were determined for the 0–10, 10–20, 20–30 and 30–50 cm layers of *A. senegal* plantations of varying age (ranging from 7 to 24-years-old) and adjacent grasslands (reference) at two sites in semi-arid areas of Sudan. At both sites, three plots were established in each grassland and plantation. The potential of *A. senegal* for $N_2$ fixation in relation to plantations age was assessed using $\delta^{15}N$ isotopic abundances and nutrient limitations assessed using C:N:P stoichiometry.

**Results:** Soil concentrations of all studied nutrients were relatively low but were significantly and directly correlated to SOC concentrations. SOC and nutrient concentrations were the highest in the topsoil (0–10 cm) and increased with plantations age. Acacia foliage $\delta^{15}N$ values were >6‰ and varied little with plantations age. Soil C:N and C:P ratios did not differ between grassland and plantations and only 0–10 cm layer N:P ratios showed significant differences between grassland and plantations.

**Discussion:** The results indicated that soil fertility in the Sahel region is strongly related to SOM contents and therefore highlighting the importance of trees in the

landscape. The higher mineral nutrient concentrations in the topsoil of the plantations may be an indication of 'nutrient uplift' by the deeper roots. The high foliar $\delta^{15}N$ values indicated that $N_2$ fixation was not an important contributor to soil N contents in the plantations. The accretion of soil N cannot be explained by deposition but may be related to inputs of excreted N brought into the area annually by grazing and browsing animals. The soil C:N:P stoichiometry indicated that the plantations may be limited by P and the grasslands limited by N.

# INTRODUCTION

Over the last decades sub-Saharan Africa has experienced severe land degradation and food security challenges linked to loss of soil fertility and soil organic matter (SOM), recurrent drought and increasing population (*Nkonya et al., 2015*). While soil water availability is the main limitation on primary productivity in drylands, nutrient deficiencies, particularly nitrogen (N), phosphorus (P) and potassium (K), are other important causes (*FAO, 2004*; *Lal, 2004a*). SOM plays an important role in maintaining adequate nutrients and moisture levels (*Tiessen, Cuevas & Chacon, 1994*; *Lal, 2004b*) and soil fertility management practices that increase SOM contents have been adopted in many drylands in order to enhance crop productivity (*FAO, 2004*; *Koohafkan & Stewart, 2008*). The use of a fallow period is a well-known practice in these areas, allowing the soil to restore its SOM content and so recover from years of cultivation (*Sanchez, 1999*). However, the area of land put under fallow and the duration of the fallow period have been reduced as a result of increasing population pressure (*Kaya, Hildebrand & Nair, 2000*; *FAO, 2004*). Other practices aimed at reversing land degradation have focused on the role of trees, particularly $N_2$-fixing species, in maintaining soil fertility and protecting the soil from wind and water erosion (*FAO, 2001*, *2004*). The deeper roots of trees play an important role in mineral nutrient recycling, enabling mineral nutrients to be taken up from deeper soil layers and making them available to ground vegetation via litterfall—so-called 'nutrient uplift' (*Scholes, 1990*; *Ludwig et al., 2004*).

Sub-Saharan drylands are characterized by woodland savanna with trees and shrubs forming an open canopy with varying proportions of grasses (*Bourlière & Hadley, 1983*; *Torello-Raventosa et al., 2013*). The importance of the facilitative mechanisms (relative to competition) of trees in tree-grass systems has been reported to be greater in drier savanna (*Dohn et al., 2013*; *Moustakas et al., 2013*). The positive effects of trees and shrubs on ground vegetation have been attributed to the effect of shade, improvement in soil moisture conditions, and increased nutrients contents under tree canopies (*Belsky et al., 1993*; *Hagos & Smit, 2005*; *Blaser et al., 2013*). Fire in savanna is typical, although varying in frequency and intensity, and generally results in a loss of C and N from the ecosystem (*Pellegrini et al., 2015*). However, fire may have little effect on soil total N and soil organic carbon (SOC) because of the superficial nature of the fires

(*Coetsee, Bond & February, 2010*; *Coetsee, Jacobs & Govender, 2012*). Savanna ecosystems are also subject to grazing and browsing, the effects of which on ecosystem biogeochemistry and nutrient fluxes are complex and variable, but maybe significant (*Holdo et al., 2007*). In open ecosystems, such as savannas, herbivores may bring in significant quantities of nutrients, particularly N and P, in the form of dung and urine (*Holdo et al., 2007*).

$N_2$ fixation can increase soil N contents (*Ludwig et al., 2004*; *Blaser et al., 2013*). However, $N_2$ fixation has a high P requirement (*Vitousek et al., 2002*; *Binkley, Senock & Cromack, 2003*), which is low in dryland soils due to P adsorption either by iron oxide (*Dregne, 1976*) or calcium (*Lajtha & Schlesinger, 1988*). The abundance of stable N isotopes ($\delta^{15}N$) in leaves and, to a lesser extent, soils can be used to assess $N_2$ fixation and indicate patterns of ecosystem N cycling (*Boddey et al., 2000*; *Aranibar et al., 2004*; *Peri et al., 2012*). Low foliar and soil $\delta^{15}N$ values indicate biological $N_2$ fixation (*Schulze et al., 1999*; *Robinson, 2001*), while the enrichment of soil $^{15}N$ can be attributed to SOM reprocessing by microorganisms (*Aranibar et al., 2004*; *Swap et al., 2004*).

The biogeochemical cycles of C, N and P are often closely related (*Finzi et al., 2011*) and C:N:P stoichiometry is commonly used to provide an insight into the nature of nutrient limitations in ecosystems (*Jobbágy & Jackson, 2001*; *Bui & Henderson, 2013*). Soil C:N and C:P ratios are useful indicators of the state of SOM decomposition and N and P availability (*Batjes, 1996*; *Tian et al., 2010*), and foliar N:P ratios have been used to assess plant nutrient limitations (*Ludwig et al., 2004*; *Sitters, Edwards & Olde Venterink, 2013*; *Blaser et al., 2014*).

*Acacia senegal* (L.) Willd. (the new scientific name is *Senegalia senegal* (L.) Britton.) is a highly drought-resistant tree native to Sudan and Sahel zone of Africa (*Obeid & Seif El Din, 1970*). Although the new name has been used in a number of recent publications, we have retained the use of the old name, *A. senegal*, for reasons of consistency with our previous two related articles and with literature in general, and because of the local importance of the old name. *A. senegal* provides a wide variety of benefits, such as fodder for animals, fuelwood, charcoal and gum arabic (*Barbier, 1992*). Gum arabic is an exudate collected from *A. senegal* trees and widely used as an emulsifier in confectionary and beverages, photography, pharmaceutical and other manufacturing industries (*Barbier, 2000*). This tree is also known to be capable of $N_2$ fixation under different soil types and climatic conditions (*Raddad et al., 2005*; *Gray et al., 2013*). The influence of *A. senegal* on soil physiochemical properties in arid and semi-arid areas of Africa has been documented in a number of studies (*Deans et al., 1999*; *Githae, Gachene & Njoka, 2011*). In Sudan, particular attention has been given to SOC and N contents under *A. senegal* in the north Kordofan region (*Jakubaschk, 2002*; *Olsson & Ardö, 2002*; *Ardö & Olsson, 2004*; *Abaker et al., 2016*) and on the influence of inter-cropping systems with *A. senegal* on soil properties of sandy and clay soils (*Raddad et al., 2006*; *El Tahir et al., 2009*).

The aims of our study were to determine the effects of *A. senegal* plantation age on: (1) soil N (total), P (total and available) and K (total and exchangeable) concentrations, stocks and accretion rates; (2) potential $N_2$ fixation using foliar $\delta^{15}N$ values; and (3) acacia

leaf, ground vegetation N:P ratios and soil C:N:P stoichiometry in order to indicate nutrient limitations, imbalances and cycling in these ecosystems. We hypothesized that soil N, P and K concentrations and stocks would be positively correlated with SOC and increase with planation age, further indicating the benefits of maintaining tree cover in these semi-arid environments. This paper complements two previous papers dealing with effects of *A. senegal* plantation age on SOC stocks (*Abaker et al., 2016*) and on soil moisture and water balance (*Abaker, Berninger & Starr, 2018*). These two studies were carried out at the same sites as in this study.

## MATERIALS AND METHODS

### Study sites

We conducted our research at two sites in western Sudan: El Demokeya forest reserve (13°16′ N, 30°29′ E, 560 m a.s.l.), an experimental site managed for gum arabic research, and El Hemaira forest (13°19′ N 30°10′ E, 570 m a.s.l.) owned and managed by farmers for gum arabic production (Fig. 1). At both sites there was an area of open grassland which was taken to serve as a control against which the plantations of differing age were compared. Photographs showing the plantations and grasslands at the two sites during the rainy season are given in Supplementary Material S1.

The long-term mean annual rainfall and temperature for both sites is 318 mm and 27.3 °C. The soils at both sites are classified as Cambic Arenosols (FAO) ($\geq$90% sand). The topography is very gently sloping eastwards at El Demokeya and flat at El Hemaira and the hydrology similar at the two sites. Water balance modelled runoff from the grasslands was 32 and 95 mm for 2011 and 2012 respectively, zero for the plantations in 2011 and 63 mm in 2012 at both sites (*Abaker, Berninger & Starr, 2018*). Drainage was higher in 2011 than in 2012, and somewhat less at El Hemaira (ranging from 0 to123 mm) than at El Demokeya (ranging from 25 to 128 mm). The vegetation at both sites falls within the low rainfall woodland savanna type (*Ayoub, 1998*; *FAO, 2006*). Main components of the ground vegetation at both sites were grasses such as *Cenchrus biflorus*, *Aristidia pallida* and *Eragrostis tremula*, and some herbs, including *Geigeria alata*, *Justicia kotschyi*, *Trianthema pentandra* and *Acanthus* spp. A complete list of ground vegetation species found at the two sites in given in Supplementary Material S2. Although site-specific information about grazing and frequency of fire at the two sites is unavailable, it is known that there is over-grazing by sheep and browsing by camels, even within the forest reserve at El Demokeya. Additional information about the study sites and sampling have been described in *Abaker et al. (2016)*.

### Experimental design, sample plots and sampling

The plantations were 15 and 24-years-old (in 2011) at El Demokeya and seven, 15 and 20-years-old at El Hemaira. The same experimental design was used at both sites. Three circular plots (17.8 m radius; 0.1 ha) were established in each plantation of differing age at both sites. Three square plots (50 × 50 m at El Demokeya and 30 × 30 m at El Hemaira) were located in the adjacent grassland, the difference in size being due to the difference in the area of grassland available at the two sites. Square rather than circular

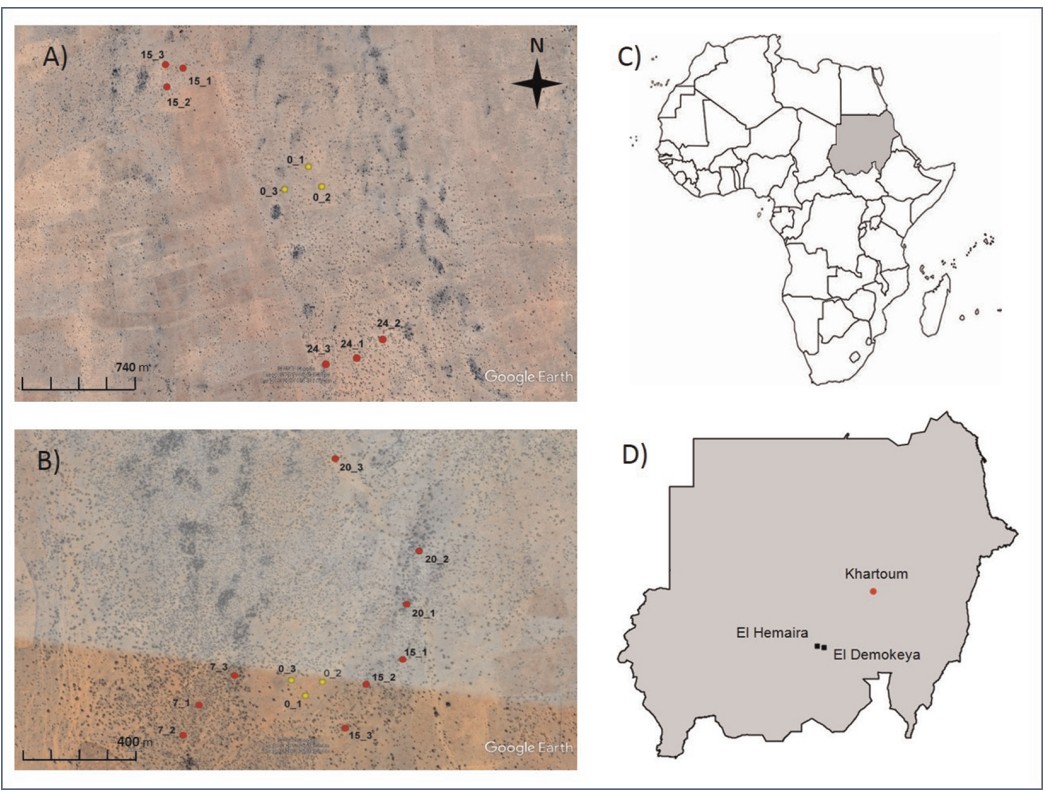

**Figure 1 Satellite images of the two study sites El Demokeya (A) and El Hemaira (B) showing location of the plots.** Number preceding the underscore refers to plantation age in years (0 = grassland) and number following the underscore refers to plot number. Inset maps showing Sudan's location in Africa (C) and location of study sites in Sudan (D). Image: © 2017 Google, DigitalGlobe and CNES/Airbus.

plots were used for the grasslands as it was easier to delineate in the field and to carry out the sampling. Although constituting a pseudoreplicated experimental design (*Hurlbert, 1984*), which limits the generality of our results concerning patterns about plantation age effects sensu stricto, the design was determined by the spatial layout of the plantations at the study sites.

Acacia leaf samples were taken from three trees (randomly selected) per plot when the foliage was fully developed. Seven randomly selected terminal branches were collected from each of the three trees, air dried, and the leaves excised and composited by tree ($n = 45$).

Ground vegetation samples were collected from one (randomly selected) of the three replicate plots per plantation age and the ground vegetation (a mixture of grasses and herbs) from 14 quadrats ($1 \times 1$ m$^2$) harvested. For the grasslands, ground vegetation samples were collected from three quadrats located in one of the grassland plots at each study site. Sampling was carried when the ground vegetation was fully developed. All the plants within each quadrat were manually uprooted, separated into above and belowground parts in the field and placed into separate bags. There were a total of 76 quadrats.

Soil samples were taken from the 0–10, 10–20, 20–30 and 30–50 cm layers of all plantation and grassland plots. For each of the plantation plots, samples were taken from

under the canopy of one (randomly) selected tree at three distances (0.5, 1 and 2.5 m) from the stem. For the grassland plots, samples were taken from the four corners and centre of each plot and composited by layer. For determination of bulk density for the grasslands, a separate sample was collected from the centre of only one of the grassland plots at each site.

## Sample pre-treatment and laboratory analysis

The tree-wise composited acacia leaf samples were further composited by plot for chemical and N isotope analyses ($n = 15$). The above and belowground vegetation biomass samples were dried at 60 °C for 48 h and weighed. However, in order to reduce analytical costs, the samples from only five of the 14 quadrats per plantation plot and two of the three quadrats from each of the grassland plots were selected (randomly) for analyses and only the aboveground samples analysed ($n = 29$). The soil samples were air-dried and passed through a two mm sieve and the <2 mm fraction saved for analysis. In the case of the soil samples from the plantations, the samples from the three distances from the stem were combined for total elemental analysis, otherwise the other analyses were carried out on the individual samples.

Contents of C and N in the acacia leaves, vegetation and soil samples were determined directly using an elemental CN analyser (Vario MAX CN; Elementar Analysensysteme GmbH, Germany). Contents of P and K were determined by digesting the samples (300 mg plant material, one g soil) in concentrated $HNO_3$ acid ($10 + 1$ ml $H_2O_2$) and microwaving, and measuring elemental concentrations using an ICP-OES spectrometer (Thermo Scientific iCAP 6000 Series, USA). Particle size analysis of the sieved soil samples was performed using a laser diffraction device (Coulter LS230; Coulter Corporation, Miami, FL, USA) and the percentage of clay, silt and sand fractions calculated. The total elemental and particle size analyses were carried out on oven-dried samples (105 °C). Soil available phosphorous ($P_{av}$) was extracted using 0.5 M sodium bicarbonate solution (pH 8.5) and concentrations determined using the Molybdenum blue spectrophotometer method and exchangeable K ($K_{ex}$) was extracted with one M ammonium acetate (pH 7.0) and concentrations determined by flame-photometer, both following FAO guidelines (*Dewis & Freitas, 1970*) and were determined from the air-dried samples. Apparent (also known as 'tapped') bulk density was determined using approximately 20 ml of soil placed into a measuring cylinder, tapped 10 times, and the volume and weight of the soil used to calculate the bulk density (*Tan, 2005*). This method is recommended because of the difficulty in taking intact volumetric samples from loose sandy soils with no structure (*Tan, 2005*), as was the case with our sites. The determination of $P_{av}$, $K_{ex}$ and bulk density was made in the laboratory of the Agricultural Research Corporation, Ministry of Agriculture, Sudan while the total elemental and particle size analyses were carried out in the laboratory of the Department of Forest Sciences, University of Helsinki.

The abundance of stable nitrogen isotope, $^{15}N$, was determined from the acacia leaf, ground vegetation aboveground biomass and soil (only for one grassland plot per site) samples. $\delta^{15}N$ values were determined using continuous-flow isotope ratio mass

spectrometry at the Centre for Stable isotopes at IMK-IFU/KIT Garmisch-Partenkirchen (Germany). The precision (standard deviation) of internal standard for stable N isotopic composition was better than 0.2‰. The stable isotopic composition of nitrogen is expressed relative to atmospheric $N_2$ (international standard for N).

## Calculation of soil stocks and accretion rates

Soil organic carbon and nutrient stocks (g m$^{-2}$) were calculated using both the traditional fixed depth method and the minimum equivalent soil mass (ESM) method (*Lee et al., 2009*). The fixed depth stocks were calculated according to the following equation:

$$Stocks = soil\,concentration \times BD \times T \times 100$$

where concentration is in %, BD is soil bulk density (g cm$^{-3}$) and T the thickness of the soil layer (cm). The ESM stocks for each layer were calculated according the equations given by *Lee et al. (2009)*. This was done in order to eliminate the effect of any alteration in bulk density associated with plantation age. The stocks for the four sampling layers were summed to give the stocks for the 0–50 cm layer. Accretion rates of nutrients in the soil were calculated as the difference between the grassland and the oldest plantation fixed depth stocks divided by the age of the plantation.

## Statistical analysis

The effect of plantation age (grassland was taken to be 0-years-old) on SOC, N, P, $P_{av}$, K and $K_{ex}$ concentrations by layer and stocks (0–50 cm) and on C:N:P ratios by layer were tested for each site separately using one-way analysis of variance followed by Tukey post hoc tests. Differences in acacia leaf N, P and K concentrations, N:P ratios and soil and acacia leaf δ$^{15}$N abundances between the 7, 15 and 20-year-old plantations at El Hemaria were similarly tested, but for El Demokeya a *t*-test was performed as there were only plantations of two ages.

The dependence of the total soil N, P, $P_{av}$, K and $K_{ex}$ on SOC was evaluated by fitting linear regressions and the coefficient of determination ($R^2$). Correlations (Pearson) between SOC contents and total N, P, $P_{av}$, K and $K_{ex}$ were computed for each soil layer and site separately. All the statistical analyses were performed using SPSS software (IBM SPSS Statistics for Windows, Version 22.0; IBM Corp., Armonk, NY, USA).

## RESULTS

All nutrient concentrations were generally higher in the plantations than in the grasslands, increased with plantations age and decreased with depth (Fig. 2). Concentrations of SOC, N, total P and $K_{ex}$ in the top (0–10 cm) layer were significantly ($p \leq 0.05$) higher in the oldest plantations at both sites compared to the grassland plots. Soil concentrations of total N, P, $P_{av}$ and $K_{ex}$ also significantly depended on SOC concentrations (Fig. 3). The strongest dependence was for N ($R^2 = 0.90$) and the weakest was for total K ($R^2 = 0.11$). The correlations between SOC and N concentrations were significant for all layers at both of the sites (Table 1). The correlations between SOC and total P

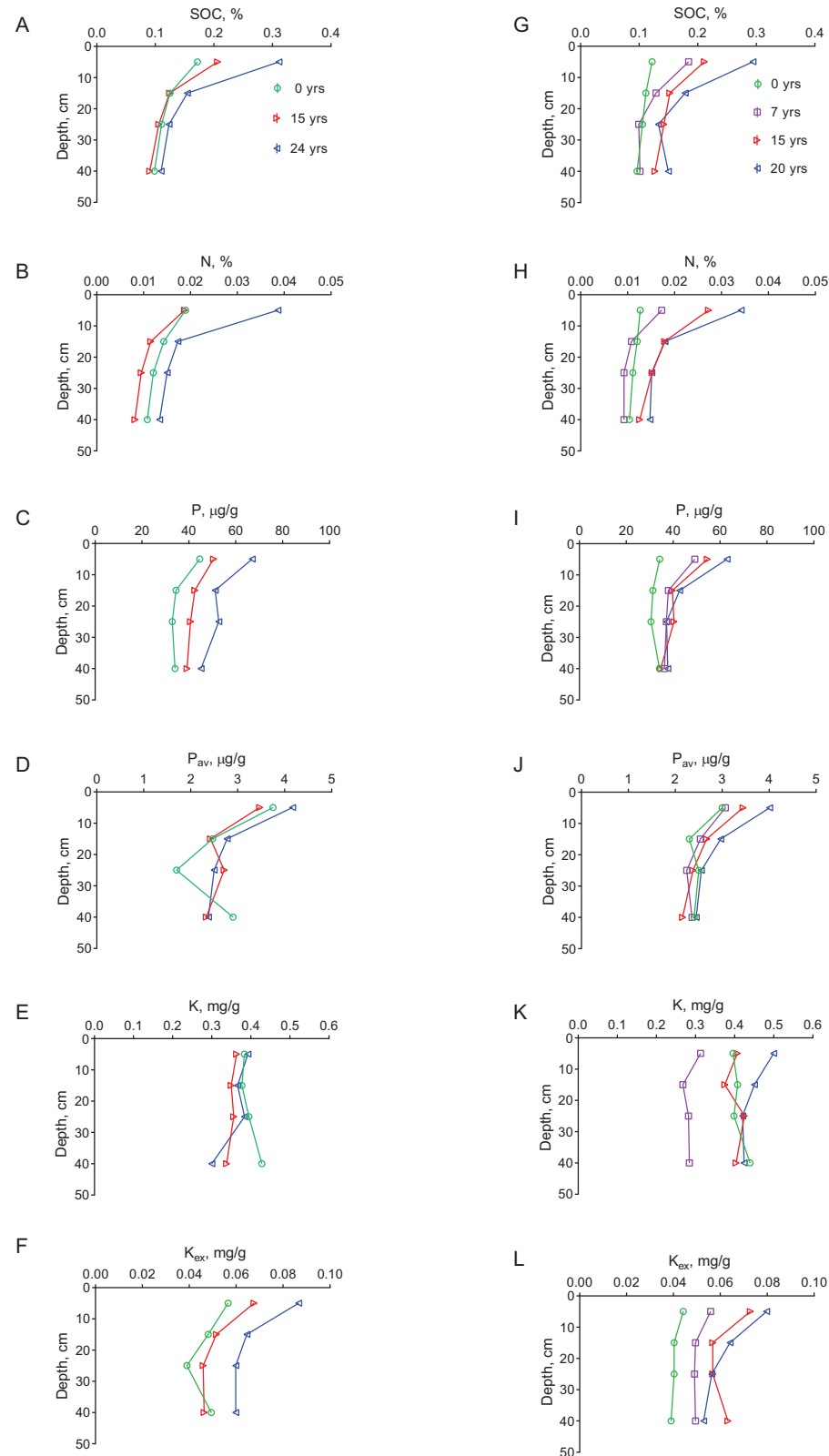

**Figure 2 Soil SOC, N, total P, available P, total K and exchangeable K mean (*n* = 3) concentrations plotted against depth for grassland and plantations by age for El Demokeya (A–F) and El Hemaira (G–L) sites.** SOC data from *Abaker et al. (2016)*.

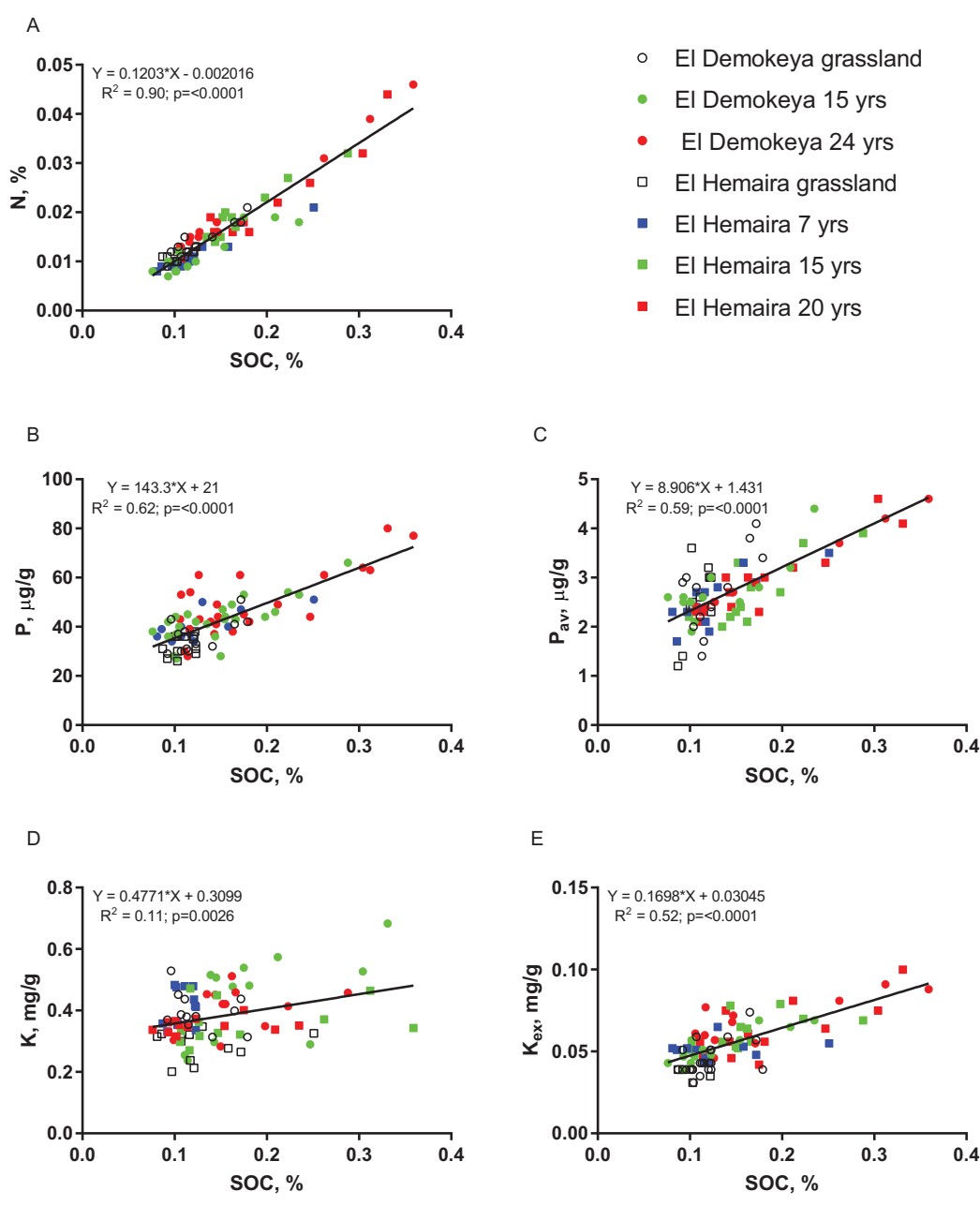

**Figure 3 Dependence of soil N (A), total P (B), available P (C), total K (D) and exchangeable K (E) on SOC concentrations for grassland and plantations by age across all soil layers and for the two study sites.**

concentrations were significant for all layers at El Hemaira but in the case of El Demokeya the correlation was significant only for the top layer. The correlations between SOC and total K concentrations were stronger at El Hemaira than at El Demokeya. In the case of $P_{av}$ and $K_{ex}$, significant correlations with SOC were associated with the upper layers.

Nutrient stocks in the soil of the plantations were generally greater than those in the grassland and increased with plantation age (Table 2). As the fixed depth SOC stock

**Table 1 Pearson correlations between SOC and N, total P, available P, total K, exchangeable K concentrations by soil layer across all plots separately for El Demokeya ($n = 9$) and El Hemaira ($n = 12$) sites.**

| Site | Layer, cm | N | P | $P_{av}$ | K | $K_{ex}$ |
|---|---|---|---|---|---|---|
| El Demokeya | 0–10 | **0.942** | **0.915** | 0.634 | 0.006 | **0.749** |
| | 10–20 | **0.675** | 0.600 | **0.817** | −0.323 | 0.637 |
| | 20–30 | **0.652** | 0.442 | 0.144 | −0.064 | 0.366 |
| | 30–50 | **0.729** | 0.182 | −0.302 | −0.307 | **0.757** |
| El Hemaira | 0–10 | **0.950** | **0.869** | **0.848** | 0.566 | **0.762** |
| | 10–20 | **0.827** | **0.699** | **0.657** | 0.558 | **0.862** |
| | 20–30 | **0.906** | **0.732** | 0.434 | **0.732** | 0.529 |
| | 30–50 | **0.936** | **0.663** | 0.170 | **0.576** | 0.365 |

Note:
   Significant ($\alpha = 0.05$) correlations are given in bold.

**Table 2 Soil stocks (g m$^{-2}$; 0–50 cm layer) of SOC, N, total P, available P, total K and exchangeable K for grassland and plantations (under canopy) by age for the two study sites.**

| Site | Age | SOC* | N | P | $P_{av}$ | K | $K_{ex}$ |
|---|---|---|---|---|---|---|---|
| El Demokeya | 0** | 950(51)[a] | 105 (11)[a] | 28 (3.1)[a] | 2.1 (0.1)[a] | 315 (33)[a] | 38.0 (8.0)[a] |
| | 15 | 1024(143)[ab] | 93 (10)[a] | 35 (1.5)[ab] | 2.2 (0.2)[a] | 291 (5)[a] | 43.0 (4.5)[a] |
| | 24 | 1260(122)[b] | 153 (15)[b] | 41 (7.6)[b] | 2.2 (0.1)[a] | 273 (28)[a] | 51.5 (2.1)[b] |
| El Hemaira | 0** | 867(59)[a] | 92 (1)[a] | 27 (2.0)[a] | 2.1 (0.6)[a] | 339 (43)[a] | 33.0 (0.6)[a] |
| | 7 | 982(190)[ab] | 89 (13)[a] | 32 (1.2)[a] | 2.0 (0.3)[a] | 230 (40)[a] | 40.6 (3.1)[ab] |
| | 15 | 1216(138)[ab] | 136 (27)[ab] | 33 (6.4)[a] | 2.0 (0.2)[a] | 323 (60)[a] | 50.1 (2.0)[b] |
| | 20 | 1422(240)[b] | 151 (32)[b] | 34 (6.4)[a] | 2.3 (0.3)[a] | 349 (119)[a] | 48.1 (9.6)[b] |

Notes:
   Values are mean values ($n = 3$) followed by standard deviation (in parentheses). Within each site, mean values sharing the same superscript letters (a, ab, b) are not significantly different from each other (Tukey's HSD, $\alpha < 0.05$).
   * SOC values from *Abaker et al. (2016)*.
   ** Grassland.

values showed better relationships with SOC concentrations and with plantation age than did ESM SOC stock values, only the fixed depth stock SOC and nutrient values are presented in Table 2 and handled further. However, the ESM SOC and nutrient stock values are presented in Supplementary Material S3. At El Demokeya SOC, N, total P and $K_{ex}$ stocks were significantly higher in the oldest plantation than those in the grassland, but not $P_{av}$ and total K stocks. At El Hemaira SOC, N and $K_{ex}$ stocks were also significantly higher in the oldest plantation than in the grassland. $K_{ex}$ stocks in the 15-year-old plantation were also significantly higher than in the grassland. Assuming that the significant difference between grassland and the oldest plantation SOC, N and total P stocks represents the addition of these elements brought about by the effect of the plantation, the average under canopy accretion rates of SOC and N at El Demokeya would be respectively 12.9 and 2.0 g m$^{-2}$ yr$^{-1}$. At El Hemaira, the corresponding SOC and N accretion rates would be 27.8 and 3.0 g m$^{-2}$ yr$^{-1}$. The total P accretion rate at El Demokeya

**Table 3 Soil C, N and P stoichiometric ratios for the grassland and plantations by age and layer (cm). Values are plot age mean values ($n = 3$).**

| Site | Age (years) | C:N | | | | N:P | | | | C:P | | | |
|---|---|---|---|---|---|---|---|---|---|---|---|---|---|
| | | 0–10 | 10–20 | 20–30 | 30–50 | 0–10 | 10–20 | 20–30 | 30–50 | 0–10 | 10–20 | 20–30 | 30–50 |
| El Demokeya | 0* | 9.1 | 8.8 | 9.2 | 9.2 | 4.3[a] | 4.2 | 3.7 | 3.2 | 38.9 | 36.7 | 34.0 | 29.7 |
| | 15 | 11.1 | 10.9 | 11.2 | 11.2 | 3.7[a] | 2.7 | 2.4 | 2.1 | 41.0 | 29.2 | 26.1 | 23.1 |
| | 24 | 8.1 | 8.9 | 8.2 | 8.2 | 5.7[b] | 3.4 | 2.9 | 3.0 | 46.4 | 30.4 | 23.9 | 24.8 |
| El Hemaira | 0* | 9.6 | 9.2 | 9.5 | 9.3 | 3.8[a] | 3.9 | 3.7 | 3.1 | 36.2 | 35.6 | 35.1 | 28.2 |
| | 7 | 10.5 | 11.9 | 10.7 | 11.1 | 3.5[a] | 2.9 | 2.5 | 2.5 | 37.4 | 34.1 | 26.8 | 28.0 |
| | 15 | 8.7 | 8.7 | 9.5 | 10.3 | 5.0[ab] | 4.6 | 3.7 | 3.6 | 43.4 | 40.4 | 35.4 | 36.8 |
| | 20 | 8.8 | 10.0 | 9.0 | 10.3 | 5.5[b] | 4.2 | 4.1 | 3.9 | 48.1 | 41.5 | 36.3 | 39.6 |

Notes:
Values within each site and soil layer sharing the same superscript letter (a, ab, b) or having no letter are not significantly different from each other (Tukey's HSD, $\alpha < 0.05$).
* Grassland.

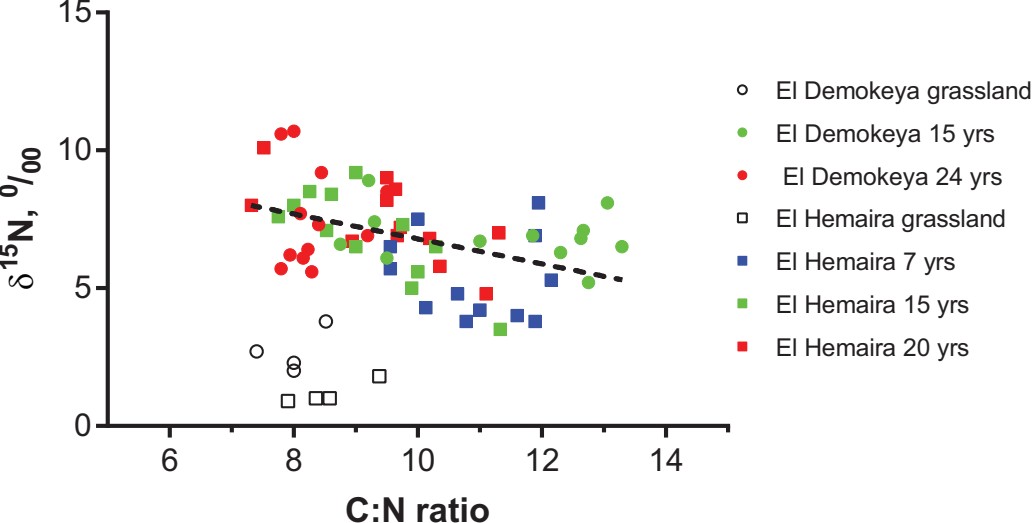

**Figure 4 Relationship between soil $\delta^{15}$N (‰) and soil C:N ratios for grasslands and plantations by age for the two study sites across all plots and layers.** Dashed line is the linear regression fitted to the plantation data only ($Y = -0.452^*X + 11.31$, $R^2 = 0.1926$, $p = 0.0005$).

would be 0.5 g m$^{-2}$ yr$^{-1}$ (as the difference in total P stocks between the grassland and oldest plantation at El Hemaira was not significant, the accretion rate is considered zero).

Grassland ground vegetation and soil $\delta^{15}$N values were generally lower than corresponding plantation $\delta^{15}$N values. Acacia foliar $\delta^{15}$N values were higher than ground vegetation values, but neither showed a difference related to plantation age (Table 3). The number of ground vegetation samples was too small to allow for significance testing. Soil $\delta^{15}$N values increased with plantation age and decreased with depth at both study sites, but these trends were not significant ($p > 0.05$). Plantation soil $\delta^{15}$N values were significantly correlated to soil C:N ratios, but the relationship for grasslands was clearly different (Fig. 4).

**Table 4 N, P and K concentrations (mg g$^{-1}$) and N:P ratios in acacia leaves ($n = 3$), aboveground vegetation in the grassland ($n = 2$) and plantations ($n = 5$) at each of the two study sites.**

| Site | Age (years) | Sample | N | P | K | N:P |
|---|---|---|---|---|---|---|
| El Demokeya | 0* | Grd. veg. | 12.1 | 2.7 | 23.5 | 4.6 |
| | 15 | Acacia leaves | 40.0 (1.8) | 0.7 (0.05) | 4.6 (0.1) | 59.3 |
| | | Grd. veg. | 11.6 (1.6) | 2.3 (0.6) | 19.9 (3.5) | 5.3 |
| | 24 | Acacia leaves | 39.4 (1.7) | 0.7 (0.04) | 4.2 (0.6) | 59.2 |
| | | Grd. veg. | 13.0 (1.6) | 1.8 (0.3) | 16.6 (5.8) | 7.6 |
| El Hemaira | 0* | Grd. veg. | 11.8 | 0.9 | 17.1 | 13.5 |
| | 7 | Acacia leaves | 38.4 (2.9) | 0.5 (0.0) | 5.2 (1.3) | 70.3 |
| | | Grd. veg. | 16.1 (4.5) | 1.5 (0.3) | 16.6 (8.0) | 11.1 |
| | 15 | Acacia leaves | 41.4 (2.1) | 0.6 (0.0) | 3.5 (0.5) | 73.7 |
| | | Grd. veg. | 11.6 (3.7) | 1.8 (0.6) | 20.3 (4.6) | 6.9 |
| | 20 | Acacia leaves | 40.0 (5.3) | 0.9 (0.4) | 4.4 (0.8) | 52.6 |
| | | Grd. veg. | 19.2 (5.3) | 3.9 (0.7) | 30.7 (8.3) | 4.9 |

**Note:**
Values are means followed by standard deviation (in parentheses).
* Grassland.

**Table 5 $\delta^{15}$N values (‰) for acacia leaves ($n = 3$), aboveground vegetation ($n = 2$ for grassland, and $n = 5$ for plantations) and soil ($n = 3$) by plantation age at the two study sites.**

| Site | Age (years) | Acacia leaves | Ground veg. | Soil layer (cm) | | | |
|---|---|---|---|---|---|---|---|
| | | | | 0–10 | 10–20 | 20–30 | 30–50 |
| El Demokeya | 0* | – | 2.9 | 3.8 | 2.7 | 2.3 | 2.0 |
| | 15 | 6.5 (1.5) | 3.2 (1.1) | 7.9 (1.1) | 6.9 (0.2) | 6.8 (0.6) | 5.9 (0.7) |
| | 24 | 7.0 (1.5) | 3.8 (1.5) | 10.2 (0.9) | 7.7 (0.8) | 6.4 (0.8) | 6.0 (0.4) |
| El Hemaira | 0* | – | 5.8 | 1.8 | 1.0 | 1.0 | 0.9 |
| | 7 | 8.8 (1.2) | 7.9 (2.0) | 7.4 (0.8) | 5.4 (1.3) | 4.5 (1.1) | 4.3 (0.5) |
| | 15 | 8.9 (0.5) | 5.5 (1.7) | 8.7 (0.4) | 7.1 (1.3) | 6.5 (1.2) | 5.5 (1.7) |
| | 20 | 8.0 (1.4) | 6.7 (1.2) | 9.1 (0.9) | 7.4 (1.0) | 6.9 (1.1) | 6.2 (1.3) |

**Notes:**
Values are mean values followed by standard deviation (in parentheses). Soil grassland value is for a single composite sample from one plot.
* Grassland.

Soil C:N:P ratios did not significantly differ with depth and the N:P ratios only showed significant differences between plantation age for the 0–10 cm layer (Table 4). The 0–10 cm soil layer C:N and C:P ratios did not show significant differences with age at either of the sites. At El Demokeya, the 0–10 m soil layer N:P ratio in the 24-year-old plantation was significantly ($p < 0.05$) higher than those in the grassland and 15-year-old plantation. At El Hemaira, the 0–10 cm soil layer N:P ratio in 20-year-old plantation was significantly greater ($p < 0.05$) than those in the grassland and 7-year-old plantation. Acacia leaf nutrient concentrations and N:P ratios did not show significant differences related to plantations age at either of the sites (Table 5). There were too few ground vegetation samples for statistical testing of nutrient concentrations and ratios.

## DISCUSSION

In this study we aimed to determine whether the previously reported increase in SOC contents with plantation age at the two sites (*Abaker et al., 2016*) would also result in higher nutrient (N, P and K) concentrations and stocks, which would further support the importance of maintaining or increasing tree cover in the region. In an earlier paper, we showed that the increases in SOC with plantation age at the two sites resulted in increased available water capacities which then had an effect on the water balance of the plantations (*Abaker, Berninger & Starr, 2018*). Because of the pseudoreplicated design of our study, general patterns about plantation age effects may not be strictly inferred. However, given the inevitable within site variation in site conditions, the climate, soil type and topography were uniform across each site and the replicate three plots for each treatment (grassland and plantation age) were located so as otherwise to be as similar and comparable as possible. Unfortunately, documented information about land-use prior to the establishment of the plantations at the two sites was not available. However, from discussions with local staff, the *A. senegal* plantations were established on areas of homogenous abandoned grassland.

Recognising the potential limitations imposed by the pseudoreplicated design, the significant dependence of nutrient concentrations on SOC and the significantly higher N and $K_{ex}$ stocks in the oldest plantations compared to the grasslands support our initial hypothesis that soil N, P and K are linked to SOC and are in agreement with results reported from other studies. For example, in *A. tortilis* savanna woodlands in northern Tanzania *Ludwig et al. (2004)* found increases in SOM, N, P and $P_{av}$ concentrations with tree growth stage (grassland, under small and large trees), and *Deans et al. (1999)* working with *A. senegal* in Senegal found that N and $K_{ex}$, but not P concentrations, increased with plantation age. In both these studies, the soil refers to the surface layer (0–10 cm). This layer had the highest SOC contents and would therefore be expected to be the most affected by the plantations. Furthermore, in the study by *Deans et al. (1999)*, soil concentrations of N, P and $K_{ex}$ were all significantly correlated to loss-on-ignition contents, i.e. SOC contents. *El Tahir et al. (2009)* working at El Demokeya site, reported a SOC stock value of 738 g m$^{-2}$ for 0–30 cm layer and for total N, $P_{av}$ and $K_{ex}$ values of 118, 2.5 and 29 g m$^{-2}$, respectively. We were unable to take into account the effect of fire and grazing on soil SOC and N stocks at our study sites. However, fire has generally been found not to result in a loss of soil total N and SOC because of the superficial nature of the fires (*Coetsee, Bond & February, 2010*; *Coetsee, Jacobs & Govender, 2012*). The effect of grazing at our study sites is discussed below in relation to soil N stocks.

Compared to the grasslands, the higher N, $P_{av}$ and $K_{ex}$ concentrations observed in the upper soil layer of the plantations indicates a significant effect of acacia trees on ecosystem nutrient cycling, at least at our study sites. The higher concentrations in the surface layer was particularly obvious in the older plantations and can be explained by 'nutrient uplift' by the deeper roots of the acacia trees (*Scholes, 1990*; *Ludwig et al., 2004*). *Mubarak, Abdalla & Nortcliff (2012)* also concluded that tree litter input is a significant source of P and K in southern Kordofan soils and the presence of trees has been

shown to contribute to the general maintenance of soil fertility in the Sahel (*Wezel, Rajot & Herbrig, 2000*; *Schlecht et al., 2006*).

The higher N concentrations in the surface soils of the plantations may be thought to be due to $N_2$ fixation as acacia species are considered to be $N_2$ fixing (*Ludwig et al., 2004*; *Raddad et al., 2005*; *Boutton & Liao, 2010*). Although, *A. senegal* has been reported to be a $N_2$ fixer (*Raddad et al., 2005*; *Isaac et al., 2011*; *Githae et al., 2013*; *Gray et al., 2013*) the high $\delta^{15}N$ values we observed for acacia leaves (>6‰) would indicate that *A. senegal* did not fix $N_2$ or is very limited in our sites. If there had been significant $N_2$ fixation in the plantations then one would expect foliar $\delta^{15}N$ values to be closer to 0‰ (*Robinson, 2001*; *Aranibar et al., 2004*; *Nardoto et al., 2014*). Nevertheless, our acacia foliage $\delta^{15}N$ values are in agreement with the findings of other studies conducted in arid environments. For example, *Aranibar et al. (2004)* observed that Acacia leaves had $\delta^{15}N$ values similar to non-legume species and even higher than known $N_2$-fixing species in a study carried in the Kalahari Desert. *Pate et al. (1998)* reported a mean $\delta^{15}N$ value of 9.10‰ for Acacia species in arid Australia, which was identical to those of non-fixing woody species, suggesting little or absence of N fixation. In a study carried out in *A. tortilis* savanna woodlands in Kenya, *Belsky et al. (1993)* concluded that that $N_2$ fixation was not an important contributor of N to the soil. $N_2$ fixation by legume trees in drylands has been show to vary considerably, even within the same species (*Nygren et al., 2012*). For example, $N_2$ fixation by *A. senegal* growing on clay soil in Sudan was shown to vary from 29 to 48 kg N ha$^{-1}$ (*Raddad et al., 2005*).

Our soil N accretion rates in the plantations appear high but are comparable to those reported by *Blaser et al. (2014)* of 1.3–2.0 g N m$^{-2}$ yr$^{-1}$ (for 0–10 cm layer) in Zambian savanna. However, the vegetation at their site was dominated by the $N_2$-fixing shrub *Dichrostachys cinerea*. As deposition loads of N in the Sahel are about 0.3–0.7 g N m$^{-2}$ yr$^{-1}$ (*Delon et al., 2010*), our high N accretion rates cannot be explained by deposition. The paradox between the accumulation of soil N in the absence of $N_2$ fixation and sufficient N deposition in humid tropical forests has been identified in several studies (see *Hedin et al., 2009*) and has been explained by heterotrophic $N_2$ fixation by free-living bacteria decomposing litter and SOM (*Vitousek & Hobbie, 2000*) or by canopy epiphytic $N_2$ fixation (*Hedin et al., 2009*). However, the rates of such $N_2$ fixation are low and could not explain our high soil N accretion rates. A possible source of our observed high soil N accretion rates could be from grazing animal excretion. The two study sites are not fenced and seasonal pastoral and nomadic grazing (mainly sheep) and browsing (camels), although varying, takes place throughout the study area (*Poussart, Ardö & Olsson, 2004*). Bigger trees (older plantations) may be expected to provide increased shading and ground vegetation for grazing and browsing. Animals entering the plantations may therefore have added N to the soil in the form of animal excretion derived from grazing outside and in excess of grazing removals from inside the study sites. Studies on elk and bison in north-temperate grassland indicate that herbivore excretion can add significant amounts of N to the soil (*Frank et al., 1994*). However, data on land-use history and animal herbivory at the two sites is not available and therefore this animal excretion N explanation is only speculative.

It has been shown that N-fixing trees accumulate large amount of N-rich litterfall during the first years of establishment, however once N availability has built up in the soil, N fixation may be ceased or inhibited (*Khanna, 1998*; *Boddey et al., 2000*; *Hedin et al., 2009*) and the older trees/plantations become more dependent on litterfall and N recycling (*Deans et al., 2003*). The relatively high and increasing trend in soil $\delta^{15}$N with plantation age at our sites may be an indication of greater microbiological processing of SOM and a more open N cycle (ammonia volatilization and denitrification during the wet season) resulting in an enrichment of $^{15}$N (*Aranibar et al., 2004*; *Swap et al., 2004*; *Hobbie & Ouimette, 2009*). The negative relationship observed between soil $\delta^{15}$N and soil C:N ratios in the plantations is consistent with the notion that low soil C:N ratios in arid environments promote greater N gaseous losses (*Austin & Vitousek, 1998*; *Aranibar et al., 2004*; *Saiz et al., 2016*). The vegetation present at a given site exerts a large influence on SOM dynamics not only because of the quantity and quality of organic matter returning to the soil (*Saiz et al., 2015*), but also because of its impact on soil hydrological conditions (*Abaker, Berninger & Starr, 2018*). In this regard, trees growing on coarse-textured soils in semi-arid regions may promote the maintenance of soil water conditions suitable for the activities of SOM decomposers through the interception and funnelling of rainfall by their canopies and the reduction in soil water evaporation by shading (*Bargués Tobella et al., 2014*; *Ilstedt et al., 2016*). Two recent works have shown potentially faster SOM decomposition rates at locations dominated by trees compared to those dominated by grass vegetation in mixed $C_3/C_4$ systems occurring on coarse-textured soils (*Saiz et al., 2015*; *2016*). These vegetation-related factors may be responsible for the higher SOC and nutrient contents observed in our acacia plantations. The higher soil $\delta^{15}$N values observed with plantation age is further evidence of SOM decomposition processes being comparatively more dynamic under the direct influence of trees.

Cyanobacteria associated with the formation of cryptogamic soil crusts have been shown to be a significant pathway to fix atmospheric $N_2$ in arid environments, but their development diminishes with vegetation cover (*Aranibar et al., 2004*; *Wang et al., 2013*). Therefore, N fixation by cyanobacterial soil crusts (which may be expected to be more strongly developed in the grasslands) may explain the low soil $\delta^{15}$N values observed in grassland sites. However, we have no information on the presence and development of such cyanobacterial soil crusts at our sites, but in any case annual N fixation rates associated with cyanobacterial soil crusts are very low (*Aranibar et al., 2003*).

The decreasing rather than increasing trend in soil $\delta^{15}$N with depth observed in both the grasslands and plantations is somewhat unusual (*Hobbie & Ouimette, 2009*), but it has also been observed in an arid, sandy site in the Kalahari (*Wang et al., 2013*). The variation in soil $\delta^{15}$N values with depth are the result of multiple interacting factors, which include N inputs by plant and cryptogamic crusts, vertical transport processes (i.e. leaching, fungal immobilization and bioturbation), soil moisture conditions, and isotopically fractionating processes (e.g. ammonia volatilization and denitrification) (*Hobbie & Ouimette, 2009*; *Wang et al., 2013*; *Saiz et al., 2016*). However, as the N contents in our soils are very low resulting in a low analytical signal for $^{15}$N, our soil $\delta^{15}$N results should be interpreted with caution.

Soil C:N ratios often decrease with soil depth as a result of the SOM being older and more decomposed and therefore relatively enriched in N compared to SOC (*Batjes, 1996*; *Tian et al., 2010*). However, there was no consistent trend in C:N ratios with depth in either the grasslands or the plantations at our sites, which may be explained by gaseous losses of N as indicated by the soil $\delta^{15}N$ values and discussed above. The significantly lower soil (0–10 cm) N:P ratios in the grasslands than in the oldest plantations at our sites would indicate N limitation in the grasslands. The N:P ratios of the ground vegetation were on the lower side of values presented for savanna grasses by *Ludwig et al. (2004)* and *Sitters, Edwards & Olde Venterink (2013)*. *Ludwig et al. (2004)* considered low grass N:P ratios from open grasslands to indicate N limitation and higher values for grasses sampled from under the canopy of trees to indicate P-limiting conditions for the grasses. *Sitters, Edwards & Olde Venterink (2013)* similarly concluded that their observed increase in grass N:P ratios with tree density indicated a shift towards P-limiting conditions for the ground vegetation.

## CONCLUSION

The concentrations of all studied nutrients were relatively low but directly and significantly correlated to SOC, were highest in the topsoil and increased with plantation age at our sites. Although these results are specific to our study sites, we consider these results support our hypothesis that soil N, P and K contents in the Sahel region are strongly controlled by SOM (SOC) contents. Although *A. senegal* is known to be capable of $N_2$ fixation and may have occurred when the trees were young, current foliar $\delta^{15}N$ values did not indicate ongoing $N_2$ fixation in the plantations. The soil N accretion rates observed in the plantations were unlikely to be due to N deposition but may be related to inputs of excreted N brought into the area annually by grazing and browsing animals. The relatively high surface soil N contents in the plantations at our sites were considered to be the result of litterfall and recycling. The higher total and plant available contents of P and K in the soil surface of the plantations may be an indication of 'nutrient uplift' by the deeper roots of the acacia trees. Soil N:P ratios indicated N limitation in the grasslands and a trend towards P-limitation in the plantations. Our results support the notion that an increase in SOM (SOC) contents related to the retention and preferably planting of trees in the Sahel region would not only increase carbon sequestration, but also significantly improve soil fertility.

## ACKNOWLEDGEMENTS

We would like to thank the staff of El Obeid Agricultural Research Station, Sudan for their logistic support and help with the fieldwork. We appreciate the technical assistance by laboratory technician Marjut Wallner during the analysis of the samples at the Department of Forest Sciences, University of Helsinki. We also thank Mr Victor Braojos for his assistance with the isotope analysis.

### Funding

This work was supported by the Academy of Finland project, Carbon Sequestration and Soil Fertility on African Drylands, (CASFAD) and grant from University of Helsinki.

The funders had no role in study design, data collection and analysis, decision to publish, or preparation of the manuscript.

## Grant Disclosures

The following grant information was disclosed by the authors:
Academy of Finland project, Carbon Sequestration and Soil Fertility on African Drylands, (CASFAD) and grant from University of Helsinki.

## Competing Interests

Frank Berninger is an Academic Editor for PeerJ.

## Author Contributions

- Wafa E. Abaker conceived and designed the experiments, performed the experiments, analysed the data, contributed reagents/materials/analysis tools, prepared figures and/or tables, authored or reviewed drafts of the paper, approved the final draft.
- Frank Berninger conceived and designed the experiments, authored or reviewed drafts of the paper, approved the final draft.
- Gustavo Saiz contributed reagents/materials/analysis tools, authored or reviewed drafts of the paper, approved the final draft.
- Jukka Pumpanen authored or reviewed drafts of the paper, approved the final draft.
- Mike Starr conceived and designed the experiments, analysed the data, prepared figures and/or tables, authored or reviewed drafts of the paper, approved the final draft.

## Data Availability

The raw data are provided in a Supplemental File.

## Supplemental Information

Supplemental information for this article can be found online at http://dx.doi.org/10.7717/peerj.5232#supplemental-information.

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
