# Peer review of "Linkages between soil carbon, soil fertility and nitrogen fixation in Acacia senegal plantations of varying age in Sudan"

_PeerJ, doi:10.7717/peerj.5232_

## Round 0.1 · original submission · Major Revisions

Although the three reviewers have recognized that your manuscript has merits, at least two identified major deficiencies which should be addressed before the ms. can be published. In particular, I want to stress the following:
NOVELTY:
* Reviewers 2 and 3 noticed the similarity between the present manuscript and previous paper by the same authors (Abaker et al 2016 and Abaker et al. in press). If parts of Figure 3 and Table 2 are identical to SOC results reported in the former paper, these results should be excluded and simply referred to in the discussion only, removing the emphasis in the current manuscript on the SOC results. More generally, the novelty/added value of the present ms. should be presented very clearly, and already published data recognized as such.
METHODOLOGY:
* As raised by reviewer 1, the experiment is pseudo-replicated (plots for each treatment located together spatially); this needs to be acknowledged, i.e. you need to discuss the possibility that the changes in SOC and nutrients with age of plantation could have been inherited from the sites and acknowledge this in the interpretation of the results.
* If soil samples were actually dried at 105 degrees prior to analysis of organic matter, it is needed to discuss the implications for the results (see reviewer 2).
* The laboratory packing method for bulk soil density assessment should be detailed, and an adequate reference given. Similarly, more details on the root sampling method should be given.
* I would like you to seriously address the concern of reviewer 2 that comparison of soil nutrient and carbon stocks between different land uses may be expressed in terms of soil mass in addition to soil volume.
* Use the right taxonomic name for the tree species (you can indicate what was the old name).
DATA INTERPRETATION:
* Reviewer 1 also identified kind of self-contradiction in the following reasoning: (i) you argue that there has been negligible N fixation, but (ii) as the plantations age, the N cycle becomes more open with enhance N volatilization. So (iii) how does N increase with time, and how the increasing N is coming from N leaf litter cycling if it is not initially derived from N fixation (what would be the source then)? The reviewer encourages you to use previous papers for addressing this paradox.
OTHER MANUSCRIPT IMPROVEMENTS:
* Keep the order of the main issues addressed in this study consistent between the introduction; methods and results.
* I also recommend that you seriously consider the recomendation of referee 3 to provide a summary table of soil carbon stocks and rates of carbon sequestration in relation to other studies.

The reviewers have made a range of other suggestions that will help you to improve the ms.

When you will submit the revised version, please carefully address each point raised by the reviewers in your response letter and manuscript.

best regards

Xavier LE ROUX

·

Basic reporting

This paper is written in clear professional English. It is of a good standard with adequate referencing, and professional handling of data, tables and figures.

Experimental design

The experiment is pseudo-replicated, in that the plots for each treatment are located together spatially. This limits the applicabilitity of the findings. The problem is not fatal, but needs to be acknowledged in the interpretation of the results. The experimental layout leaves open the possibility that the differences (increases) in SOC and nutrients with age of plantation could have been inherited from the sites. This possibility needs to be discussed.

Validity of the findings

In Lines 260 - 276, the authors argue that there has been no N fixation, but as the plantations age, the N cycle becomes more open with enhance N volatilization. So how does N increase with time. This query also applies to the Conclusions in section 5. If N fixation is not occurring, how can you argue that the increasing N is coming from N cycling in leaf litter? Where is the N ultimately coming from? It was low to start with - so what was the source? If it was leaves going into leaf litter, then where ultimately was the source for the leaves? Could there be something wrong with the logic. Have a read of Cook and Dawes-Gromadzki 2005 Landscape Ecology 20: 649-660 who looked at some of these issues in an Australian acacia landscape. This paradox needs to be addressed. At the moment it is self-contradictory.

Additional comments

Line 118: Eragrostis (no "e")

·

Basic reporting

I have no major comments with regards to basic reporting (although I have included some minor concerns in the general comments below). The manuscript was fairly clear and generally well structured.

Experimental design

See comments under general with regard to methodology queries

Validity of the findings

See general comments below

Additional comments

The manuscript by Abaker et al. describes nutrient (N, P, K) and soil carbon stocks in the soil and vegetation under grassland and Acacia senegal plantations of different ages. This study reports data from a poorly studied region with some very pressing environmental and land management issues. The authors have provided a fairly clear and comprehensive discussion of their dataset and the implications of plantation establishment for recovering soil fertility.
I have a number of concerns with some of the methodology descriptions and analyses that I have presented below.
1. Soil sampling and analyses:
The authors mention that soil samples were dried prior to analysis at 105 degrees. This is unfortunate, as it is likely that some organic matter was volatilised at this high temperature (see Reuter, D.J., Robinson, J.B., Peverill, K.I., Price, G.H. and Lambert, M.J. (1997). Guidelines for collecting, handling and analysing plant materials. In: ‘Plant Analysis: An Interpretation Manual, 2nd Edition’, pp. 53-70, (D.J. Reuter and J.B. Robinson Eds) (CSIRO Publishing: Collingwood)). It might be helpful to discuss the implications of this for your results.
Secondly, the authors mention on line 153-154 that the apparent bulk density of composited soil samples was determined using the ‘laboratory packing method’. I am unfamiliar with this method and I would like to see a more detailed explanation of what it entails. My main concern is that by compositing soil samples much of the sample structure has been lost, making accurate bulk density measurements quite difficult. This may then have a substantial impact on your nutrient stock calculations presented in Table 1. However, I may be mis-interpreting this paragraph and perhaps bulk density determinations were performed on individual intact soil samples?
My final comment is in regard to comparison of soil nutrient and carbon stocks between different land uses that may vary considerably in terms of bulk density. When SOC and STN stocks are being compared across sites where bulk density varies and soils are sampled to a fixed depth, SOC and N stocks will need to be re-calculated to an equivalent soil mass before comparisons are valid, otherwise differences between sites in nutrient content may be solely caused by differences in volume of soil sampled. Re-calculation of SOC and STN stocks to an equivalent soil mass (usually calculated from the site with the lowest bulk density) is performed by determining the relationship between cumulative soil mass and cumulative C or N stock at each site (see Gifford and Roderick 2003; Lee et al. 2009). It might be helpful to discuss this potential issue in your dataset, particularly if there was substantial soil impacts or compaction during plantation establishment.
Gifford RM, Roderick ML (2003) Soil carbon stocks and bulk density: spatial or cumulative mass coordinates as a basis of expression? Global Change Biology 9, 1507–1514
Lee J, Hopmans JW, Rolston DE, Baer SG, Six J (2009) Determining soil carbon stock changes; Simple bulk density corrections fail. Agriculture, Ecosystems and Environment 134, 251-256.
2. Sample types
The authors mention that above and belowground samples of ground vegetation were taken and analysed for nutrient content. I was curious as to how root samples were removed – eg. was the main root ball dug up, was it rinsed prior to dying etc.? There was also an indication on line 146 that these samples were analysed separately, but only a single value was reported in Table 4- is this a combination of below and aboveground biomass samples or just aboveground samples? If below ground samples were combined with aboveground, it is unclear then why root nutrient contents were not sampled for the trees as well.

My other concern was the similarity between this manuscript and a previous manuscript by Abaker et al (2016) that also reports soil organic carbon (SOC) stocks and contents under the same plantations. Parts of Figure 3 and Table 2 are identical to SOC results reported in this publication, and this should perhaps be noted or these results excluded and referred to in the discussion only. This would also remove any emphasis in the current manuscript on the SOC results, as the implications of this dataset have already been discussed and published elsewhere.

A number of minor concerns are presented below:
Abstract line 43: the ‘nutrient uplift’ effect was not directly measured in this manuscript so it may be best to rephrase this sentence as “We speculate that the higher mineral nutrient concentrations….”
Introduction line 71: the wording of this sentence is a little unclear in terms of how SOM contents are related to increased nutrient contents and cycling under tree canopies (it would seem that these processes result in higher SOM contents rather than the other way around).
Introduction line 78: It might be helpful to start a new paragraph focused on N fixation here.
Introduction line 104: change hypothesized to hypothesize
Results line 186: The reference to Fig. 2 should be removed as this does not show nutrient concentrations with depth and age
Results line 190: Remove “as could be expected”
Discussion line 235 to 238: Reference and re-iteration of the results should be removed.
Discussion line 246-255 and 340-342: It might be helpful to indicate that you speculate that ‘nutrient uplift’ may be the mechanism enhancing nutrient contents in surface soils
Discussion line 346: I was not entirely clear how improving soil fertility under Acacia plantations would benefit the local community. Is it because they will then be cleared and used for food production? Or is it because it improves the diversity and palatability of ground vegetation that is used for grazing? It might be helpful to either remove this sentence or place the context of plantations and soil fertility and benefits to local communities into the introduction.
Figure 1: It would be helpful to include a map of the study sites that includes their location in reference to the whole of Sudan (similar to the Abaker et al. 2016 reference).
Figure 3: Would it be possible to include error bars here
Table 1 to 5: Could you include number of replicates (where applicable) for values where standard deviations have been calculated in the Table labels e.g. “Mean soil stocks (g m-2; 0-50cm ± (standard deviation), n = 3)…”
Table 2: the formatting of the superscript letters in the SOC column is a little confusing.
Best wishes in addressing these concerns,
Dr Anna Richards

·

Basic reporting

This study builds on Abaker et al. (2016) — which investigated biomass and soil carbon in various aged plantations of Acacia senegal — be exploring potential co-correlates with important plant nutrients.
This is a very well-written and well-structured article that clearly explains the importance of the research. Figures are relevant, correctly labelled etc. There is one issue that the authors need address: contextualising the other studies on the same system (i.e. Abaker et al. 2016 and Abaker et al. In Press). There are hints throughout the manuscript about these studies, but the reader is expected to piece them together to figure out that these studies share the same study system, and I assume the same research program. Please tell the reader up front in the introduction that this study system has been explored along different avenues and summarise those avenues. That helps contextualise this contribution in relation to the other two. (I continuously had this nagging feeling that the other two studies may have looked at some other area).
There is an issue with the species name used: Acacia senegal. The new scientific name for this species is Senegalia senegal (L.) Britton. Although I strongly dislike the manner in which the name change for African Acacia occurred, we do need to maintain the correct taxonomic links. Officially, I should ask the authors to change the name throughout, but I think they could make a convincing argument that the old name has local importance and association (e.g. with government officials?) and so the old name is more accessible. If the authors can provide a convincing argument for using the old name, then the new name does need to be included in the abstract and text at some point, else the new name should be used throughout.
My last major comment is that keeping track of samples sizes from the methods to the results is tiresome and difficult. Please include the sample sizes in the tables (even if this is simply in the table heading).
A difficult comment to implement, but there seemed to be much repetition between the results text and tables, so this could be shortened. Please ensure only major main trends are reported in the results text.
In terms of maintaining a predictable order for the reader throughout, the introduction sets up the aims (e.g. L98-100), but the methods and results don’t follow this order (e.g. 148-164). It is minor, but keeping the order consistent helps the reader.
Minor comments:
Line 111: please explain what is “gum arabic”?
L118: “Eragrosties” to “Eragrostis”
Line 152: Please include model and manufacturer for the spectrometer.
Lines 148-159: These methods are not referenced. One I am not familiar with so I think providing a reference for all is necessary.
Line 164: I’ve used carbon and oxygen isotopes before, and these are referenced against international standards. Is there an international standard for nitrogen (atmosphere, maybe?)? Please include.
Please consider including an appendix figure with ground-level photos of the different sites to give international readers further insight into the study system.
Line 319: replace “simply” with “by process of elimination”. (Yes?).
Line 364: “of of”
Figure 3. Please include information about replicates.

Experimental design

The experimental design is sufficient to answer the questions raised, and is sufficiently described for experimental reproduction. Although I don’t believe it is important, the shape and size of plots varied across different treatments (e.g. grass vs plantation). Please could a brief explanation be included as to why this was done, and also any potential (or not) influence this might have had. Again, I don’t think this is important, but the change in design was odd.

Line 204: Although unlikely in such a dry environment, fire can influence SOM rates. Please briefly mention whether fires occur in these landscapes (at least to rule this out as influencing factor, as it does play a role in more mesic environments). If fires do occur, then this needs to be considered…

Validity of the findings

The interpretation is consistent with the results, i.e. the findings are valid. No further comments required.

Additional comments

This was a well-thought out study and well put-together. One way to make your results accessible to a general reader is to provide a summary table of soil carbon stocks and rates of carbon sequestration in relation to other studies. This does not need to be a comprehensive list, but a table with stocks, rates, vegetation type and mean annual precipitation will allow the reader to immediately gauge your values in comparison to other systems. One such study that is at the same MAP is Mills&Cowling (2010; J. Arid Environments, 74: 93-100), but there are many more that can be easily harvested from Google Scholar. This little additional effort will help the reader, and likely increase the citation rate for the article.

---

## Round 0.2 · Major Revisions

You have already addressed many issues that had to be addressed, and have thus improved substantially the ms., which is acknowledged by the reviewers. However, there are 2 remaining, major issues still pending:

* The experiment, even if performed at 2 locations, corresponds to a pseudo-replication design, and you have to accept that this is an issue which should be acknowledged and discussed. Thus:

(1) Be clear on this issue in the text. For instance, you can state in the Mat&Met section : "In the present study, a pseudo-replication experimental design was used. This design limits the generality of our results, i.e., we do not test here general patterns about plantation age effects sensu stricto, but rather test patterns observed at the particular sites studied (i.e., two locations only)." (the wording is indicative of course)

(2) In the discussion, you must address the issue and discuss to what extent starting conditions may have biased the results, possibly jeopardizing the conclusions ; or whether you have some evidence that initial conditions were the same. In particular, are the positions of the plots along the catena the same? Do you have historical data on land uses for the different plots ? If not, acknowledge the possible confounding factors and thus be much more cautious when concluding.

* Regarding nutrient stock calculations. Reviewer 2 indicates that time restrictions for re-computation and search of consistency of results with previous publications are not good reasons for not performing the complementary calculations required by the reviewer, and I concur. Thus:

(3) Please make these additional, complementary calculations. You could present them as a supplementary material if they are consistent with and do not alter the results currently derived from the initial calculations. If results are not consistent, you would have to also present the complementary computations and take them into account when concluding.

Would you amend the ms. taking into account these last, two important issues, I would be happy to receive the revised version which I hope would be then likely acceptable for publication. Please understand that the objectives of the reviewers and myself are to make sure that your paper presents the results of your work in the best possible way.

·

Basic reporting

The authors need to address the issue of pseudo-replication in their design.

Experimental design

The experiment is pseudo-replicated. The authors disagree that this is an issue and chose not to address it. I believe it still is and have raised a number of ways it could affect the results. The fact that studies were done at two locations is irrelevant to the argument because they are not included in a formal analysis. The authors must address the issue and justify their analysis with evidence that starting conditions did not bias their analyses.
If the authors cannot address this issue in some way, the paper should be rejected.

Validity of the findings

With pseudo replication, one can only infer that a trend may be the result of a treatment. It is not conclusive. The way the results are presented is not valid. The analyses can legitimately say that site x is different to site y, but not that it is due to the age difference of the plantations. It could be due to landscape position, prior landuse or several other possibilities.

·

Basic reporting

No comment

Experimental design

No comment

Validity of the findings

The authors have addressed a number of my concerns raised in my previous review of their manuscript titled "Linkages between soil carbon, soil fertility and nitrogen fixation in Acacia senegal plantations of varying age in Sudan". However, I still have one outstanding issue that I would like to draw attention to. In my previous review I mentioned an issue of comparing nutrient stocks measured to a fixed depth across sites with different soil bulk densities. The authors chose not to address this concern but to refer to this issue in the discussion.

I am afraid, however, that I would like to bring this point up again. I have looked at the underlying dataset for the manuscript and, while bulk density is not highly variable between sites, there are still differences that could impact overall stock calculations and the significance of differences in nutrient stocks between sites reported in Table 2. The authors mention that time restrictions and consistency of results with previous publications was their reason for not performing these calculations. However, the time to do these calculations is not onerous (1 day perhaps) and the authors are already reporting results from previous publications in this manuscript so there is little need to refer to the previous published work (with respect to nutrient stocks). If the authors could show that they had performed these calculations and that there were no differences in site nutrient stocks compared to the fixed depth calculation then I would be happy to accept that Table 2 remains unchanged. However, without this evidence it is difficult to support the statement made by the authors that the fixed depth values are sufficiently accurate. I am sorry to labour this point and hope that it can be resolved relatively easily.

Best wishes,
Dr Anna Richards

---

## Round 0.3 · accepted · Accept

Dear authors,

You have properly addressed the last reviewers' comments. I am thus happy to accept your manuscript for publication in PeerJ, and hope that you appreciated the review process and its capacity to help you to improve the manuscript.

best regards
Xavier LE ROUX

# ·

Basic reporting

No comment

Experimental design

The limitations of the experimental design have been acknowledged appropriately and the interpretation of the results takes this into account.

Validity of the findings

The interpretation of the findings takes appropriate account of the limitations of the design.

Additional comments

.

·

Basic reporting

no comment

Experimental design

no comment

Validity of the findings

no comment